# Self-Restoring Capacitive Pressure Sensor Based on Three-Dimensional Porous Structure and Shape Memory Polymer

**DOI:** 10.3390/polym13050824

**Published:** 2021-03-08

**Authors:** Byunggeon Park, Young Jung, Jong Soo Ko, Jinhyoung Park, Hanchul Cho

**Affiliations:** 1Precision Mechanical Process and Control R&D Group, Korea Institute of Industrial Technology, 42-7, Baegyang-daero 804beon-gil, Sasang-gu, Busan 46938, Korea; bgpark91@kitech.re.kr (B.P.); young89@kitech.re.kr (Y.J.); 2Graduate School of Mechanical Engineering, Pusan National University, Busandaehak-ro 63beon-gil, Geumjeong-gu, Busan 46241, Korea; mems@pusan.ac.kr; 3School of Mechatronics Engineering, Korea University of Technology & Education, 600, Chungjeol-ro, Byeongcheon-myeon, Dongnam-gu, Chungcheongnam-do, Cheonan-si 31253, Korea

**Keywords:** pressure sensors, capacitive pressure sensor, shape memory polymer, wide pressure range, porous structure, self-restoring, durability

## Abstract

Highly flexible and compressible porous polyurethane (PU) structures have effectively been applied in capacitive pressure sensors because of the good elastic properties of the PU structures. However, PU porous structure-based pressure sensors have been limited in practical applications owing to their low durability during pressure cycling. Herein, we report a flexible pressure sensor based on a three-dimensional porous structure with notable durability at a compressive pressure of 500 kPa facilitated by the use of a shape memory polymer (SMP). The SMP porous structure was fabricated using a sugar templating process and capillary effect. The use of the SMP resulted in the maintenance of the sensing performance for 100 cycles at 500 kPa; the SMP can restore its original shape within 30 s of heating at 80 °C. The pressure sensor based on the SMP exhibited a higher sensitivity of 0.0223 kPa^−1^ than a typical PU-based sensor and displayed excellent sensing performance in terms of stability, response time, and hysteresis. Additionally, the proposed sensor was used to detect shoe insole pressures in real time and exhibited remarkable durability and motion differentiation.

## 1. Introduction

With the increasing demand for wearable devices, pressure sensors that are flexible and sensitive have attracted tremendous attention. Highly sensitive and flexible pressure sensors have great potential for various applications, such as in real time health monitoring [1,2,3,4,5], speech recognition [6], electronic skin engineering [7,8,9,10,11], and soft robotics [12,13,14]. Elastomers, such as polyurethane (PU), polydimethylsiloxane (PDMS), and Ecoflex, are promising materials for these purposes because of their high compressibility, flexibility, and biocompatibility [15,16]. To improve the performance of pressure sensors, recent research efforts have been devoted to the use of microstructure layers to change the mechanical properties of thin elastomer films [17]. Microstructure layers, such as pyramids, domes, and squares, can improve sensor performance in terms of increasing various sensing parameters, such as sensitivity and response time in the low-pressure range. However, although the low viscoelastic properties of the microstructure layer allow high sensitivity and fast response time, the sensing range of such microstructure-layer-based sensors is narrow, thereby impeding their practical applications.

Recently developed wearable devices with applications in daily life require a pressure range from 0–10 kPa (soft touch) to 10–100 kPa (hard touch) [2,18]. To facilitate effective application over this range, flexible pressure sensors must be capable of detecting and distinguishing pressures over 100 kPa. Polymeric three-dimensional (3D) structures or foams are possible candidates for use in pressure sensors that are highly sensitive and have a wide detection range because of their structural deformation capabilities and mechanical compressibility [19]. To widen the pressure range and maintain high sensitivity, multiple studies have introduced flexible pressure sensors based on porous structures. Vandeparre et al. developed a capacitive pressure sensor using a flexible PU foam as dielectric films [20]. They showed that sensitivity could be tuned by adjusting the foam density, which caused the sensor to be applicable for a pressure range of 1–100 kPa. Lee et al. also demonstrated a flexible wide-range pressure sensor based on a PU foam coated with carbon nanotubes (CNT) [21]. The sensor was capable of detecting 400 kPa applied pressure and showed high repeatability and durability at a repeated pressure of 360 kPa. However, the surface of a porous structure can be easily fractured or cracked owing to the low durability of PU. These cracked surfaces degrade the sensing performance under cyclic load conditions, which limits its practical applications. The existing PU-based flexible pressure sensors do not have sufficient durability over a wide pressure range (up to ~ 1 MPa) [22,23].

Herein, we report highly flexible and compressible pressure sensors based on 3D and PU porous structures by using a shape memory polymer (SMP) [24]. The sensor demonstrated durability over a wide pressure range. The original shape of the SMP can be restored by heating for 30 s at 80 °C. The use of the SMP as the dielectric layer in capacitive pressure sensors resulted in sensors with high sensitivity and durability. The proposed sensors exhibited sensing performance of 0.0223 kPa^−1^ at 180 kPa and a high pressure tolerance (up to ~1 MPa). The sensors also showed higher durability for 100 cycles at 500 kPa with negligible hysteresis than sensor based on the typical PU porous structure. Furthermore, the applicability of these sensors in a shoe insole system was demonstrated, indicating that the SMP-based sensor is suitable for applications involving high pressure.

## 2. Materials and Methods

### 2.1. Fabrication of the PU Porous Structure and Capactive Pressrue Sensor

The fabrication process of the PU porous structure by the sugar templating process and polymer capillary effect is shown in Figure 1. SMP and clear-flex 95 (CF95) porous structures were fabricated using the sugar templating process. The sugar templating process is a simple, inexpensive, and eco-friendly means of obtaining a porous structure [25,26,27,28,29,30]. Using sugar templating process, we can easily fabricate the porous structure sensor which has high flexibility and sensor performance [19,25,28,29,30]. A 15.6 mm × 15.6 mm × 10 mm sugar cube (BEKSUL, Seoul, Korea) as a mold was prepared and placed on a Petri dish. The resin and hardener for the SMP, which is commercially available (MP-2510, SMP Technologies Inc., Tokyo, Japan), were placed in a vacuum oven at 80 °C under 10 × 10^−1^ Torr for over 1 h to remove moisture and bubbles. Then, they were mixed in a 1:1 weight ratio by using a planetary mixer (KK-250S, Kurabo, Osaka, Japan) at revolution rate of 1350 rpm and rotation of 800 rpm for 30 s. As the pot lifetime of MP-2510 is as short as 5 min, the available mixing time is limited. In addition, a CF95 precursor, which is a mixture of the resin and hardener at a 1.5:1 weight ratio, was prepared using the planetary mixer, before it was placed in a vacuum chamber under 10 × 10^−1^ Torr for 10 min. The prepared precursor was poured into a Petri dish with sugar cubes to fill any gaps within the sugar cubes that may have arisen from the capillary effect. The polymer was cured in an oven at 70 °C for 2 h, then the sugar was dissolved in water overnight. Then, the 3D porous structure was placed in an oven for 2 h at 80 °C to dry the water that remained inside the structure. An indium tin oxide (ITO)-coated polyethylene terephthalate (PET) film (Fine Chemical Industry, Seoul, Korea) was fixed on the top and bottom of the structure with a silver paste (P-100, Jin chem, Hwaseong, Korea), enabling the structure to be applied as the dielectric layer in a capacitive sensor.

### 2.2. Evaluation of the SMP Porous Structure Sensor

The system utilized to measure the mechanical properties and sensing performance of the capacitive pressure sensor is shown in Figure 2. The sensor based on the PU porous structure was positioned within a specially designed chamber, then placed on a hot plate (IKA C-MAG7, IKA-Werke GmbH & Co. KG, Staufen, Germany). The chamber helps maintain the required temperature (Appendix A). The pressure universal test machine (AGS-X, Shimadzu, Japan), with a load cell, was used (maximum load of 1 kN, resolution of 2 N) at a compression speed of 10 mm min^−1^. The capacitance was measured with an LCR meter (HIOKI-3536, HIOKI, Nagano, Japan) at 500 kHz and 1 V bias. The sensor evaluation data were acquired in real time by a computer connected to the LCR meter. A thermal image of the insole evaluation was captured using a thermal image camera (A700-EST, FLIR, Wilsonville, OR, USA)).

## 3. Results

SMP is a special PU material wherein the original shape can be restored by heating at a specific temperature, exhibiting a behavior similar to that of a shape memory alloy. It is anticipated that this property can improve the durability of the PU porous structure and, hence, maintain sensor performance. In particular, MP-2510, used in this study, has a low glass transition temperature (25 °C); thus, it can be applied to various fields, such as wearable devices designed for human motion monitoring. CF95, which is a typical polyurethane, serves as a suitable material for a control group because it has hardness and tensile strength similar to those of MP-2510. Specifically, MP-2510 exhibits hardness of shore 30 D (≈shore 80 A) and tensile strength of 20 MPa at temperatures above the glass transition temperature. The hardness and tensile strength values of CF95 are 95 A and ≈17.2 MPa, respectively.

Figure 3 shows the results for the mechanical and electrical durability of the PU porous structure-based sensors. To verify the durability of the PU porous structure-based sensors, compressive pressure of 500 kPa was applied for 100 cycles at 40 °C. For SMP, the stress–strain curves were almost identical, with negligible change during the 100 cycles (Figure 3a). In contrast, the strain values for the CF95 porous structure gradually increased as the pressure was repeated because of its low durability (Figure 3b). Fatigue fracture affects the porous structure of CF95 and produces the strain shift. In fact, the CF95 porous structure was partially destroyed after the 100-cycle test (photo inset Figure 3b). The photo images and scanning electron microscopy (SEM) images, shown in Appendix A, showed no noticeable effects on the SMP structure after the 100-cycle load test, but cracks and tears were evident in the CF95 structure. Figure 3c,d show the capacitance values for the SMP and CF95, respectively, under 500 kPa for 100 cycles. For capacitance measurement, the PU porous structure was used as the dielectric layer in the flexible capacitive pressure sensor. The SMP porous structure sensor showed consistent capacitance values for the 100-cycle pressure test, increasing by only 1.8% during the test. However, the relative capacitance of the CF95 porous structure sensor increased gradually, and was 13.5% higher than the initial value after the 100-cycle pressure test. The capacitance can be calculated using
*C* = *εA*/*d*(1)
where *ε* is the dielectric constant, *A* is the area of overlap of the two plates, and *d* is the gap between the plates. Thus, the increase in capacitance is attributed to the decrease in the distance between the electrodes during the cyclic pressure test as a result of structural damage. The reduced capacitance change seen in the SMP confirms the durability of the structure.

To investigate the recovery properties of the SMP over the glass transition temperature, we evaluated the response time of the PU porous structure-based sensor. A pressure of 100 kPa was applied to and then removed from the pressure sensor, and the capacitance values were recorded (Figure 4a). At room temperature (R.T., 21.9 °C, below the glass transition temperature), the SMP-based sensor showed a long response time of 146 s after the pressure was released. However, at 40 °C, the sensor showed a relatively stable and fast response time of 35 s, with negligible hysteresis. Additionally, the sensor performance at over 40 °C is shown in Appendix A. This is attributable to the restorative properties of the SMP-based sensor. It can, therefore, be anticipated that the response characteristics and sensitivity of the sensor are improved at 40 °C, and the sensor could be applied to systems operating at 40 °C. Additionally, Figure 4b shows the response times for SMP- and CF95-based sensors at two temperatures (R.T. and 40 °C) under changing strain conditions. Below 30% strain, the capacitance values of the sensors at the two temperatures were identical. However, after 30% strain, the difference increased; at 40 °C, the SMP-based sensor showed a fast response time and high sensitivity. The sensitivity can be changed by varying the compressibility, initial porosity, and dielectric constant [6]. In the case of Figure 4b, the compressibility is the same because all sensors were compressed under the same strain. Additionally, the initial porosity was also assumed to remain constant as only one type of sugar cube was used. Therefore, the change in the dielectric constant causes the sensitivity change. After 30% strain, the difference in the capacitance values at R.T and 40 °C was much larger for the SMP sensor than for the CF95 sensor. Thus, the sensing properties of the SMP can be improved by increasing the temperature to 40 °C. Figure 4c shows the capacitance values of the SMP-based porous pressure sensor during applied loading and unloading (at the same stress rate) at 40 °C. The sensor showed relatively stable capacitance values under differing pressure conditions (20, 40, 100, 200, 400, and 1000 kPa). Next, the sensor sensitivity as a function of pressure was studied. The sensitivity was calculated by
*S* = *δ*(Δ*C*/*C_0_*)/*δp*(2)
where *C* and *C*_0_ denote the capacitance in the presence and absence of applied pressure, respectively, and p denotes the applied pressure. The sensitivity of the PU porous structure sensor was split into two approximately linear sections. Figure 4d shows the sensitivity of the SMP and CF95 porous structures at 40 °C. In the first section (0 to 180 kPa), the SMP porous structure showed a sensitivity of 0.0223 kPa^−1^, which is three times higher than that of the CF95 porous structure. In addition, in the second section (180 to 1000 kPa), the SMP porous structure showed a sensitivity of 0.0077 kPa^−1^, which is almost 1.8 times higher than that of the CF95 porous structure. Figure 4e shows the results of the restorative test. The relative capacitance at room temperature was 0 at initial state, but it could not recover to 0 when the pressure was released, and it increased gradually during the cyclic compressive load because of slow response. In addition, as shown in the inset photo, during the 100-cycle test (loading pressure of 500 kPa), the SMP porous structure was deformed in the z-direction, which increased the capacitance. However, as seen in the third inset image and relative capacitance value of 0 after heat treatment, heat treatment for only 30 s in an 80 °C oven is sufficient to completely restore to initial state of the SMP porous structure.

The MP-2510 SMP porous structure is tolerant of a wide pressure range and is suitable for high-pressure applications. Additionally, its low glass transition temperature (25 °C) makes it applicable in various fields, such as wearable devices. In particular, certain locations at which the human body temperature (36.5 °C) can be maintained, such as inside shoes, are promising environments to apply MP-2510 SMP porous structure sensors. Accordingly, we placed an SMP porous structure sensor in a shoe insole (Figure 5a). A thermal image of the shoe before and after the investigation was taken by the thermal imaging camera. Before the experiment, the temperature inside the shoe was maintained at approximately 23 °C (room temperature = 21.9 °C). After walking for 1 min, the wearer removed the shoe, and the temperature was checked immediately. The shoe insole temperature increased to 35 °C, as shown in Figure 5b. The wearer then wore the shoe again and underwent alternative 60 s periods of walking and running (Figure 5c). The inset graphs in Figure 5c show 10 s detail for walking and running. The traces show that the fabricated sensor is able to discriminate varying types of human motion, such as standing, walking, and running.

## 4. Conclusions

In this study, we demonstrated that the durability of the PU porous structure can be improved by utilizing the restorative properties of SMP. We fabricated SMP and CF95 porous structures using a sugar templating process. The fabricated SMP (MP-2510) porous structure showed high durability. The SMP had approximately the same capacitance value after 100 cycles of loading at 500 kPa, but the capacitance value for the CF95 porous structure increased by 13.5%. The CF95 porous structure was damaged due to fatigue fracture, but the damage was not observed for the SMP porous structure. In addition, the SMP porous structure exhibited a three-fold higher sensitivity than the CF95 porous structure (0.0223 and 0.0088 kPa^−1^, respectively) at pressures below 180 kPa. Furthermore, after deformation upon cyclic pressure loading, the SMP porous structure can be restored through a 30 s heat treatment in an 80 °C oven. We applied the sensor to detect human motion by placing the SMP porous structure on a shoe insole, and it could clearly distinguish different motions, such as walking and running.

## Figures and Tables

**Figure 1 polymers-13-00824-f001:**
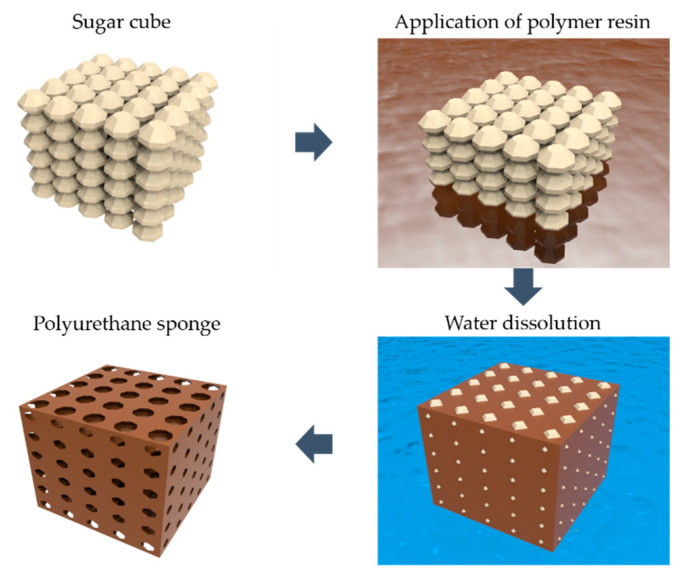
Fabrication of the polyurethane (PU) porous structure through the sugar templating process.

**Figure 2 polymers-13-00824-f002:**
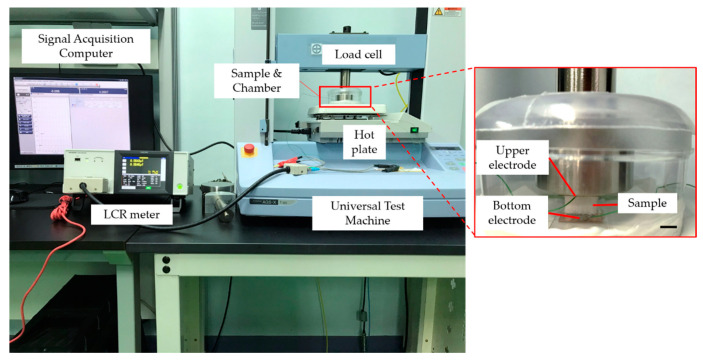
Measuring system for capacitance and mechanical properties of the PU porous structure-based sensor. Scale bar = 10 mm.

**Figure 3 polymers-13-00824-f003:**
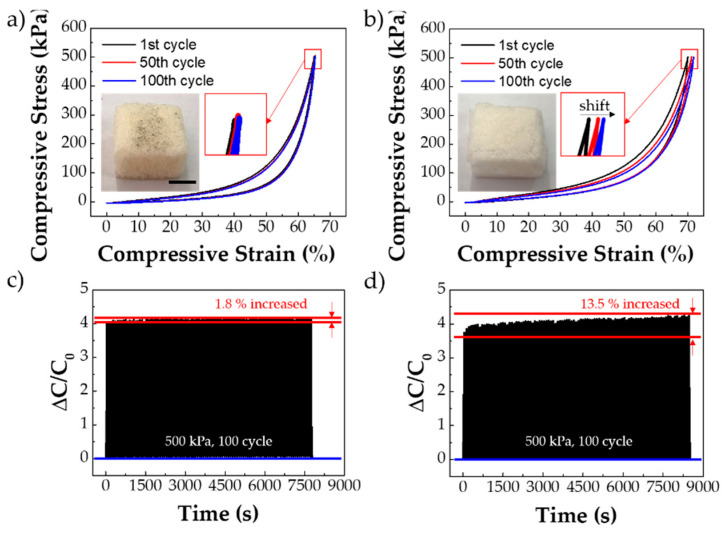
Comparison between the shape memory polymer (SMP) and CF95 porous structures. (**a**) Stress–strain curve of the SMP porous structure for 100 cycles of 500 kPa load; scale bar = 10 mm. (**b**) Stress–strain curve of the CF95 porous structure for 100 cycles of 500 kPa load. (**c**) Relative capacitance of the SMP porous structure for 100 cycle. (**d**) Relative capacitance of the CF95 porous structure for 100 cycles.

**Figure 4 polymers-13-00824-f004:**
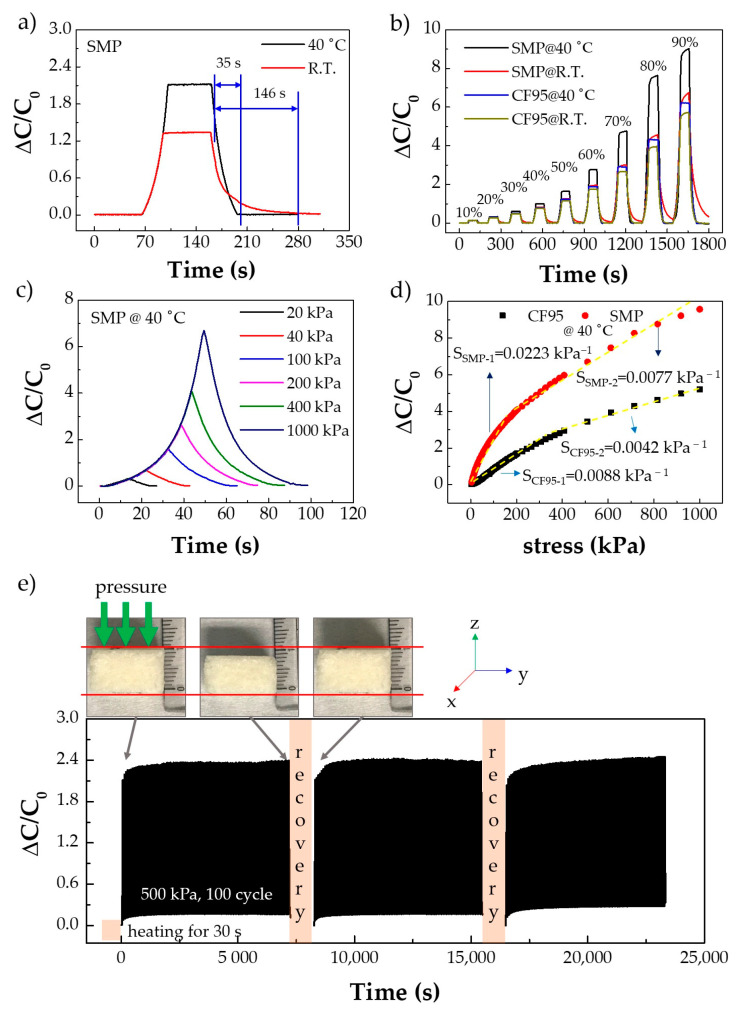
Restorative properties of the SMP porous structure and sensor performance. (**a**) Hysteresis is negligible at 40 °C due to the restorability of SMP. (**b**) Response test of the PU porous structure at various strain values. (**c**) Pressure response test of the SMP porous structure. (**d**) Sensitivity of the SMP and CF95 porous structures. (**e**) Restorability of the SMP porous structure; inset: original shape of SMP porous structure, deformed shape, and restored shape.

**Figure 5 polymers-13-00824-f005:**
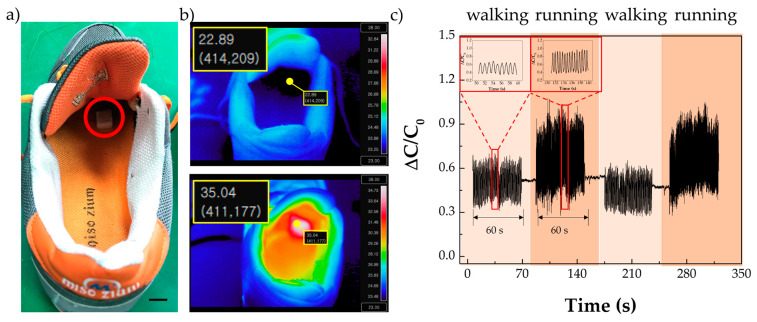
Application of the proposed sensor over a high pressure range; (**a**) SMP porous structure sensor on the shoe insole, scale bar = 20 mm; (**b**) Temperature in the shoe before the experiment and after the experiment; (**c**) Signal data of walking and running for 60 s.

## Data Availability

The data presented in this study are available on request from the corresponding author.

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
