# Peer review of "Self-Restoring Capacitive Pressure Sensor Based on Three-Dimensional Porous Structure and Shape Memory Polymer"

_polymers, 2021, doi:10.3390/polym13050824_

Round 1

Reviewer 1 Report

This paper presents a shape memory PU within porous structure through using a sugar templating process. Possessing high sensitivity and long durability, this SMP-based pressure sensor can be used to detect shoe insole pressures in real time. The design of this system is ingenious, especially the preparation method of porous structure, however, there are several problems need to be solved before this paper can be accepted and published on Polymers.

  1. As shape memory property plays an important role in the sensing performance, however, the experiments about shape memory have not been involved.
  2. As mentioned in the manuscript, the 3D porous structure and capillary effect are the key for preparing this kind of material. However, related supporting materials are missing.
  3. Actually, temperature has a great influence on the shape memory process, which can further affect the sensing property. As a result, the relationship between temperature and sensing performance should be studied.

Author Response

1. As shape memory property plays an important role in the sensing performance, however, the experiments about shape memory have not been involved.

answer) Thank you for your valuable comments. We used MP-2510, which is commercially available polymer product from SMP Tech., Japan. The characteristics of the sensor according to temperature are shown in figures 4(a) and 4(b). In addition, it can be seen from figures 4(a) and 4(b) that general PU with similar physical properties do not have such temperature-dependent properties. The recovery capacity of SMP according to temperature in the finally fabricated sensor is shown in figure 4(e). We revised the text to understand it easily (page 6, lines 197-204).

2. As mentioned in the manuscript, the 3D porous structure and capillary effect are the key for preparing this kind of material. However, related supporting materials are missing.

answer) We understood that your comment means there no enough information about sugar templating process. Sugar templating process is well-known method for making flexible porous structure. The Sugar template process is not a new process, but it is the first time to apply SMP to it. In this process we only need precursor of PU and sugar cube (page 2, lines 81-86). We add more reference about sugar templating process (reference 26~30).

3. Actually, temperature has a great influence on the shape memory process, which can further affect the sensing property. As a result, the relationship between temperature and sensing performance should be studied.

answer) We did not checked sensing performance according to temperature in detail. However, in this paper, the glass transition temperature is the most important to investigate the restore ability of the SMP. MP-2510 has glass transition temperature of 25 °C. So, we measured sensing performance under 25 °C (room temperature, 23 °C) and over 25 °C (40 °C) as shown in Figure 4(a) and (b) (page 6). Additionally, we measured sensor performance at a single cycle compressive load of 500 kPa when the temperature is 20, 40, 60, and 80 °C. We demonstrate it in Figure S3. As same as Figure 4(a) and (b), SMP porous structure cannot recover at 20 °C to initial state under the transition temperature (25 °C), but the recovery operates at 40 °C. It is same at 60, and 80 °C. Maximum capacitance value increases as temperature increases, but it just because of the change of young’s modulus according to temperature not the restore ability. We add about this in the text (page 5, lines 165-166).

Reviewer 2 Report

Existing pressure sensors based on porous polyurethane have low durability. The paper proposes to use a shape memory polymer for them. It is shown that the proposed designs have a sufficiently high sensitivity and withstand a large number of deformation cycles. In addition, when heated, they return to their original shape. In general, the proposed approach can have practical applications, so I can recommend its publication.

Author Response

Thank you for your kind comments. We are very happy to hear from you.